# OpenReview forum: "Quantifying CBRN Risk in Frontier Models"
_NeurIPS.cc/2025/Workshop/Reliable_ML — NeurIPS 2025 - Reliable ML Workshop_

### Official Review · Reviewer_fkpH · 2025-09-19
**Important experiment concerning access to high-risk specialized knowledge, lacking depth and analysis in its conclusions.**

**Rating:** 5
**Confidence:** 4

**Review:**

**Summary:** The paper evaluates whether LLMs resist CBRN (chemical, biological, radiological, nuclear) misuse. 10 models are tested (per Methods) using a 200-prompt CBRN set and a 180-prompt subset of FORTRESS. The attacks/prompts are tested on three levels of sophistication (Direct, Obfuscated, and Deep Inception). The metric used is the attack success rate. The paper observes that the most sophisticated attacks (Deep Inception) are by far the most effective on most models (with a median score of 86.0% vs 33.8% for the Direct attacks). Additionally, a large disparity across models is observed, with Claude-based models being the only ones to retain a low attack success rate throughout all different types of attacks. This suggests that proper training methods can achieve robustness in this task, but most developers have failed to do this yet.


**Strengths:** Testing of CBRN misuse risks is a highly significant task. Developers of LLMs are claiming to have tackled this; however, the findings of this paper suggest that most current safety measures are superficial/lack deeper semantic understanding. The three-tier attack taxonomy is simple and explained intuitively. Finally, sensitivities around dual-use are acknowledged with a coordinated disclosure approach and non-release of prompts/attack code.

**Weaknesses / Limitations:** The key claims of the paper (sophisticated attacks behave better, a large model disparity of 87 percentage points gap) are repeated across most sections with little additional analysis, making the paper extremely repetitive. The discussion section (section 5) provides no further insights, while also consistently referencing things mentioned already in the background and related work section. The CBRN dataset is balanced across four capability types and four domains (knowledge retrieval, process instructions, novelty, synthesis) x (Chemical, Biological, Radiological, Nuclear), but results are not reported comprehensively across the table (some results are given scattered throughout the text).

**Suggestions for Authors:** Since CBNR’s taxonomy is thoroughly discussed, one expects results per capability and domain as well. The main issue, however, is the constant repetition of the same 2-3 ideas. You should consolidate overlapping text across Results/Discussion/Conclusion, and use the space to analyze further the failures of the models, design different attacks, or suggest specific and non-vague ways developers should shield their LLMs.

**Ethics:** The topic is inherently dual-use. The paper acknowledges this and lists mitigations.

---

### Official Review · Reviewer_XAuW · 2025-09-19
**New benchmarks for evaluating the risk of frontier models generating unsafe information**

**Rating:** 8
**Confidence:** 3

**Review:**

### Summary
This work designs a benchmark to evaluate frontier models on their potential to proliferate chemical, biological, radiological, and nuclear (CBRN) weapons knowledge.  Their benchmark evaluates model responses on a variety of different aspects including their ability to recall facts, their ability to synthesize new information, and their guidance on how to synthesize weapons.  They also evaluate prompting strategies of different levels of sophistication, using direct requests, obfuscated requests, and Deep Inception, which includes levels of role-playing.  They evaluate 10 different frontier models using their new benchmark, and show that most of them can be reliably prompted to generate unsafe responses with at least one of these attacks.

### Strengths
This work provides a comprehensive benchmark of various types of attacks that could arise.  They provide a modular way to evaluate models that can be used as a benchmark for future models.  Their evaluations are thorough and complete across many aspects, including type of unsafe information, type of attack, and models that are evaluated.  The paper is very well-written and contains a thoughtful and thorough discussion of related work and design considerations, which is especially useful and important in benchmarking work, where the objectives may not be clear a priori.  It is convincing that similar techniques could be useful to evaluate models for other types of unsafe information outside of CBRN weapons knowledge.  The submission is topical for the workshop, specifically the thrust understanding the strategic manipulation of models and chatbots.

### Weaknesses/Limitations
It is possible that there are many other avenues of attacks that could be useful to evaluate, and the authors also acknowledge some potential for human subjectivity in their evaluations.  The authors are clear about the potential limitations.